# Temperature-robust activity patterns arise from coordinated axonal Sodium channel properties

**Margaret L. DeMaegd**, **Wolfgang Stein** *

School of Biological Sciences, Illinois State University, Normal, Illinois, United States of America

* wstein@ilstu.edu

**Data Availability Statement:** All relevant data are within the manuscript and its Supporting Information files. All model files are available from the ModelDB database (accession number 260972).

## Abstract

Action potentials are a key component of neuronal communication and their precise timing is critical for processes like learning, memory, and complex behaviors. Action potentials propagate through long axons to their postsynaptic partners, which requires axons not only to faithfully transfer action potentials to distant synaptic regions but also to maintain their timing. This is particularly challenging when axons differ in their morphological and physiological properties, as timing is predicted to diverge between these axons when extrinsic conditions change. It is unknown if and how diverse axons maintain timing during temperature changes that animals and humans encounter. We studied whether ambient temperature changes cause different timing in the periphery of neurons that centrally produce temperature-robust activity. In an approach combining modeling, imaging, and electrophysiology, we explored mechanisms that support timing by exposing the axons of three different neuron types from the same crustacean (*Cancer borealis*) motor circuit and involved in the same functional task to a range of physiological temperatures. We show that despite substantial differences between axons, the effects of temperature on action potential propagation were moderate and supported temperature-robust timing over long-distances. Our modeling demonstrates that to maintain timing, the underlying channel properties of these axons do not need to be temperature-insensitive or highly restricted, but coordinating the temperature sensitivities of the Sodium activation gate time constant and the maximum Sodium conductance is required. Thus, even highly temperature-sensitive ion channel properties can support temperature-robust timing between distinct neuronal types and across long distances.

## Author summary

Action potentials are electrical waves that propagate through long axons and are at the core of neuronal communication. The information they convey is encoded in their precise timing, an essential feature for learning, memory, and motor control. This timing must be maintained even though axons are exposed to various outside influences on their long journey to other brain areas and peripheral muscles. We studied how three axon types

**Funding:** This work was supported by funding from NSF 1755098 to WS. The funders had no role in study design, data collection and analysis, decision to publish, or preparation of the manuscript.

**Competing interests:** The authors have declared that no competing interests exist.

that show distinct morphological and physiological properties, but interact in a single time-sensitive behavior, respond to changes in ambient temperature. Combining imaging, electrophysiology, and modeling, we determine the mechanisms that allow axons to maintain action potential timing. We show that near temperature-insensitivity of action potential velocities supports temperature-robust timing over long-distances, but that this does not require the underlying ion channel properties to be temperature-insensitive. Indeed, ion channel properties could vary substantially in their temperature responses as long as two Sodium channel parameters—the activation gate time constant and the maximum conductance—were coordinated. Thus, even temperature-sensitive ion channels support temperature-robust action potential timing.

## Introduction

Brain circuits consist of many genetically, physiologically, and morphologically distinct neuron types with dramatically different action potential (AP) propagation velocities. Consequently, the timing of APs between these neurons changes as they travel towards the synapses [1]. Considering that most neuronal processes are time-critical, this raises the question of how appropriate timing is maintained between diverse sets of neuron types and axons. Thinner axons with lower propagation velocities within the parietal cortex, compared to those spanning across cortices, promote synchronous activity within and between brain regions [2, 3]. In invertebrates, fast and slow motor neurons have long been known to conjointly control behaviors [4]. During the squid's jet propulsion behavior, axons of different speeds are recruited to contract the entire mantle simultaneously. To achieve synchrony, motor neurons projecting more distally are larger in diameter and propagate APs faster than those projecting to proximal mantle regions [5]. Axons with distinct velocities are thus critical to maintaining the timing of APs as they propagate from action potential initiation sites to their respective axon terminals.

Yet, involving neurons with distinct AP propagation velocities poses challenges because axons display complex temporal dynamics that affect behavioral performance, as a result of activity-dependent, modulatory mechanisms [6–8], and environmental influences that alter AP propagation velocity either locally or globally. Temperature is a global perturbation that quickly affects the conductance and activation and inactivation of ion channels involved in AP generation and propagation [9]. The temperature sensitivities ($Q_{10}$s) of ion channel properties vary by several fold between and within classes of ion channels [10–12], presenting a potential problem to generate and propagate APs over a range of temperatures and more so to maintain coordinated activity between a diverse set of axons. This challenges the functioning of systems where timing is critical. In the human median nerve response, cooling the skin along the length of the nerve prolongs the latency and duration of the compound AP measured distally, and slows reflex responsiveness [13]. These effects are brought about by the distinct temperature responses of slow and fast axons that disperse AP arrival times. Considering that this is only one example of the many behaviors controlled by diverse types of neurons with varying AP propagation velocities, the question arises whether—and if so, how—the timing of propagated APs is coordinated between neurons.

Surprisingly, most explanations why behaviors are temperature-robust focus solely on AP generation. In contrast, the temperature responses of the involved axons and their importance for upholding the timing of activity established centrally has mostly been ignored. For instance, in the crustacean stomatogastric nervous system, the phase relationships of the lateral pyloric (LP), pyloric dilator (PD) and pyloric constrictor (PY) neurons are relatively constant

as a function of temperature. The relative timing of APs between these distinct neuron types appears to be maintained at the spike initiation sites through the coordination of ion channel temperature sensitivities [14]. The precision and consistency of timing across different temperatures and between individuals argues that a similar precision of activity at the muscles may be imperative for the adequate functioning of the behavior [15]. However, different neuron types propagate their component of the rhythm to the periphery individually. How temperature affects these axons and alters AP timing between neuron types is unknown.

We now show that the pyloric axons maintain phase relationships over a broad temperature range. This trait is achieved despite the substantially different propagation velocities of the LP, PD, and PY axons, corresponding to different diameters and intrinsic identities of the axons. Relatively low propagation velocity $Q_{10}$s minimized changes in timing associated with increased temperatures. Our computational approach describes generic features that enable axons such as those of the pyloric neurons to possess low propagation velocity $Q_{10}$s and support the maintenance of precise timing. We show that even highly variable channel properties result in a similarly low velocity $Q_{10}$ range, and coordination of Sodium channel property $Q_{10}$s additionally supports precise timing over long distances.

## Results

### Pyloric axons propagate APs at different velocities

We first aimed to measure the velocities of the *Cancer borealis* pyloric axons. We stimulated each axon separately at a constant temperature (10˚C, see Materials and Methods, Fig 1A and 1B). We found clear propagation velocity differences between the neuron types. On average, LP showed the fastest propagation velocity, PD was intermediate, and PY was the slowest (Fig 1C; one-way RM ANOVA, $F_{(2,8)}$ = 39.616, p<0.001, Student-Newman-Keuls *post hoc* test, P<0.01 for all comparisons, N = 5).

In unmyelinated axons, a major contributor to AP velocity is axon diameter, with larger diameters propagating faster [16]. Larger axons also produce larger AP currents, which is reflected in larger extracellular waveforms. To estimate axon diameter differences between pyloric axons, we thus first measured their amplitudes on the extracellular recordings. We compared amplitudes close to the STG, midway between the STG and peripheral muscles, and close to the muscles (Fig 1A). Fig 1E shows simultaneous original recordings of APs during three cycles of the pyloric rhythm recorded from each location. Amplitudes of LP, PY, and PD differed from one another, with LP having the largest amplitude, PD intermediate, and PY the smallest. This was consistent across recording sites and animals (Fig 1F; two-way RM ANOVA, $F_{(4,16)}$ = 6.459, p<0.01, Holm-Sidak *post hoc* tests P<0.01 for all comparisons; N = 5), suggesting that there was a comparable difference in AP currents along the length of the pyloric axons. We also found a significant positive relationship (Fig 1D; Pearson correlation coefficient, r = 0.791, p<0.001, N = 15) showing that axons with larger currents propagated APs faster.

We then wanted to confirm that larger AP currents reflected differences in axon diameters. We measured axon diameter at the *dvn* where there were clear differences in waveform amplitude. We iontophoretically injected the cell bodies of the three main pyloric neuron types, LP, PD, and PY with three different fluorescent dye colors and allowed them to diffuse throughout the neuronal processes, including the axon. Fig 1Gi,ii show the individually identifiable axons that project to the periphery through the dorsal ventricular motor nerve (*dvn*) and the maximum projection of the confocal images (see also S1 Video). The LP axon had the largest diameter (4.02μm), PD was intermediate (3.81μm), and PY axon was smallest (2.21μm). For comparison across animals, we normalized diameter measurements to the average of all three

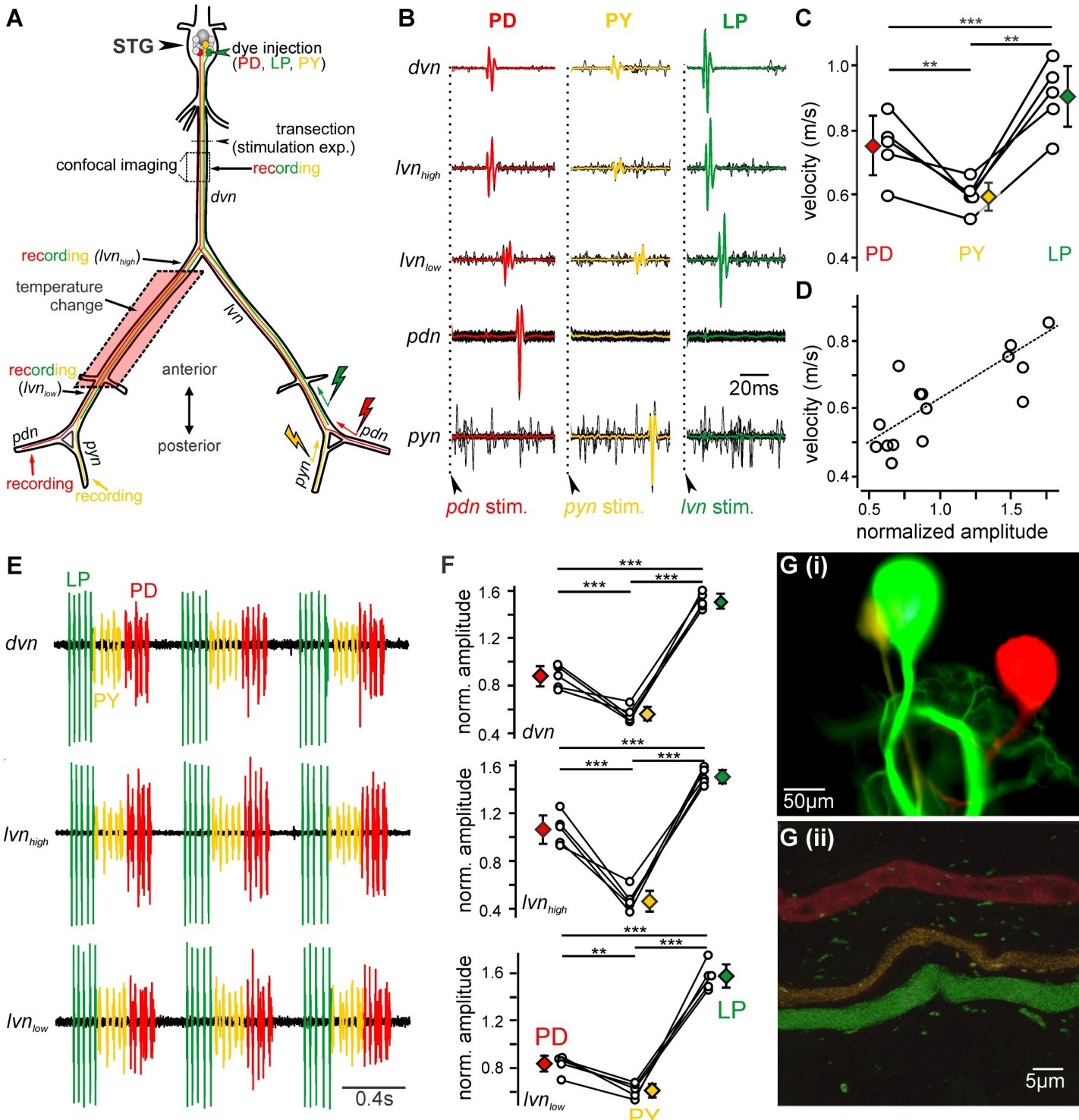

**Fig 1. Pyloric axons propagate APs at different velocities.** (A) Schematic of the stomatogastric nervous system. Nerve names are italicized. The somata and axons of the PD (red), LP (green), and PY (yellow) neurons are depicted in the color with which they were iontophoretically injected. The dotted box indicates where they were imaged. In some experiments, the *dvn* was transected, and antidromic APs were elicited in PD, LP, and PY neurons through selective stimulation of the *pdn*, *lvn*, or *pyn*, respectively, on one side of the nervous system (stimulation arrows) and propagated APs were recorded contralaterally. Colors indicate which neurons were detected on a given recording. In selected experiments, temperature was manipulated on parts of the *lvn* by bathing the nerve in temperature-controlled saline (red box). Nerves: *dvn*, dorsal ventricular nerve; *lvn*, lateral ventricular nerve; *pdn*, pyloric dilator nerve; *pyn*, pyloric constrictor nerve. (B) Overlay and average of multiple PD, PY and LP APs propagating after stimulation of the *pdn*, *pyn*, or *lvn* (as shown in A), confirming selective activation of individual APs in only one axon at a time. (C) At the control

temperature (10˚C), the pyloric axons propagate APs at significantly different velocities. (D) Propagation velocity correlates with normalized $lvn_{low}$ extracellular waveform amplitude. Amplitudes were normalized as in F. (E) Example recordings of three cycles of the pyloric rhythm from multiple locations along the *dvn* and *lvn* (compare to A) show that the extracellular AP waveform amplitudes consistently differed between neuron types. (F) Extracellular waveform amplitudes were normalized to the average amplitude of all three neurons in each experiment for comparison across animals. The normalized waveform amplitudes differed significantly from one another at all three locations. Open circles represent the waveform amplitude from a single animal. Colored diamonds represent the average for each neuron type. ** $p<0.01$, *** $p<0.001$. (G) Example image of the stained somata of the pyloric neurons (i) and maximum projection from confocal microscopy imaging used to measure axon diameter (ii).

neurons in each experiment (LP: 1.13±0.11, PD: 0.98±0.15, PY: 0.89±0.19, one-way ANOVA, F = 4.628, p = 0.021, Holm-Sidak *post hoc* test, P<0.05 for LP vs. PY, all others n.s., N≥8). Surprisingly, while the diameter differences followed the same trend, they were smaller than expected given the large differences we observed in the extracellular waveform amplitudes.

Together, our data indicate that axons with greater currents, and thus larger AP amplitudes, propagated APs faster. However, it is unlikely that axon diameter was solely responsible for the velocity differences. For equal conductances, the Hodgkin and Huxley equations predict velocities to be proportional to the square root of the diameter. From our measurements, the diameters of at least two neurons (LP and PD) show fewer relative differences than the waveform amplitudes and velocities, indicating that in these axons differences in ion channel properties may influence propagation velocity. This fits well with the fact that pyloric neurons have distinct intrinsic properties [17] and variable channel expression [18].

## Temperature robustness of axons supports functional circuit output over a wide range of temperatures

To determine whether the differences in propagation velocity affected AP timing between the neurons when temperature changes, we first recorded propagation velocity as we exposed the axons to temperatures between 5˚C and 25˚C—the range experienced by *C. borealis* in its natural habitat [19]. Propagation velocities increased for all axons as temperature increased (Fig 2Ai). While temperature responses were rather similar, they diverged starting at 19˚C, with PD showing the greatest increase, and LP and PY being significantly lower (two-way RM ANOVA, $F_{(14,56)}$ = 9.017, p<0.001, Holm-Sidak *post hoc* test 19˚C: PD-PY p<0.05, 20˚C: PD-PY and PD-LP p<0.05, 23˚C: PD-PY and PD-LP p<0.05, 25˚C: PD-PY and PD-LP p<0.05, N = 5). Fig 2Aii highlights these differences by normalizing all temperature responses to that of PD. These results were corroborated when we calculated the velocity $Q_{10}$s of the three axons by fitting a linear regression to the velocities plotted against temperature in a semilog plot (Fig 2Aiii). The linear regressions showed strong $R^2$ values and were significant (PD, $R^2$ = 0.794, p<0.001; PY, $R^2$ = 0. 881, p<0.001; LP, $R^2$ = 0.786, p<0.001). The average $Q_{10}$s were 1.53 (PD), 1.47 (PY) and 1.49 (LP). While an RM ANOVA of the linear regressions revealed significant differences in the slopes of the three axons ($F_{(2,8)}$ = 11.609, p<0.01, Student-Newman-Keuls *post hoc* test, PD-PY and PD-LP p<0.05, N = 5), their responses to temperature were surprisingly similar, given the observed differences between the pyloric axons in the extracellular recordings. With $Q_{10}$s around 1.5, pyloric axons would generally be considered temperature-robust [20, 21], which by itself would indicate a good maintenance of AP timing across different temperatures. While we did not observe any AP failures at these temperatures, others have reported that above 25˚C pyloric APs can fail [22]. We noted though that the temperature dependence of the conduction velocity did not seem to follow a single exponential but may be better described by two different regimes. In fact, a sigmoidal fit (S2 Fig) revealed an almost perfect fit for the data. This indicates that conduction velocity of all three neurons increased less at high temperatures and approached a maximum value even

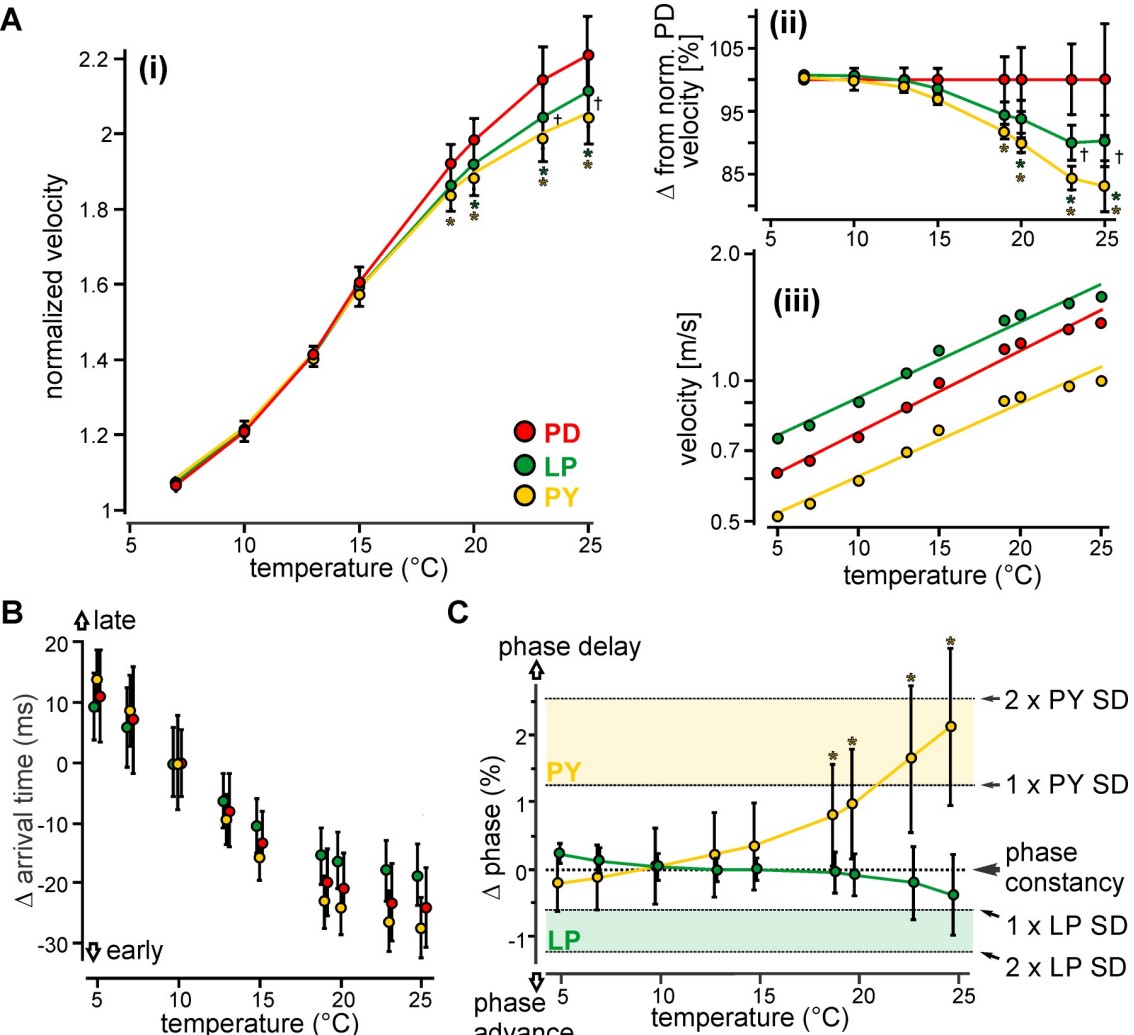

**Fig 2. Small and coordinated temperature-induced changes in propagation velocity between the pyloric axons support temperature-robust timing over long distances.** (Ai) Propagation velocity of pyloric neurons at temperatures between 5°C and 25°C. Data were normalized to the respective velocities at 5°C. (Aii) Propagation velocities of LP and PY were additionally normalized to PD to visualize the differences in propagation velocities at temperatures of 19°C and above. Asterisks denote significant differences from PD. Crosses denote significant differences between LP and PY. * $p < 0.05$, † $p < 0.05$. (Aiii) Linear regression fits of the propagation velocities of the pyloric neurons as a function of temperature. (B) Deviation in arrival time at the axon terminal (4cm from initiation sites in the STG) from the arrival time measured at 10°C of each pyloric neuron. Negative deviations indicate APs arrived earlier than at 10°C, while positive deviations indicate APs arrived later than at 10°C. (C) Percent deviation in phase near the axon terminal of LP and PY relative to the pacemaker neuron PD. Phase deviation was calculated from the change in propagation velocity across different temperatures. Phase constancy was calculated based on [14]. The standard deviations (SDs) of the individual neurons were calculated based on the variability in burst onset during spontaneous pyloric rhythms. Asterisks denote significant changes in phasing from phase constancy.

before APs failed. The flattening of the curve was more apparent for PY and LP than for PD, which is consistent with PD having the highest $Q_{10}$ value.

Neurons generally can exploit two avenues to diminish differences in AP timing at the axon terminals when temperature changes. One, their axons could be temperature insensitive ($Q_{10}$ = 1), or close to it, resulting in little to no changes of timing with temperature (S1A Fig). Alternatively, the temperature responses of the axons could be coordinated such that faster axons are less temperature-sensitive and slower axons are more temperature-sensitive (reducing the

grey bars in S1B Fig). The reasoning behind this is that proportional increases in velocity magnify differences in velocity between axons, and thus alter timing between them. The low velocity $Q_{10}$s of the pyloric axons should thus reduce timing changes between them. The subtle differences in $Q_{10}$s between PD and LP may additionally support temperature-robust timing through the alternative mechanism, because the initially faster LP axon is less temperature-sensitive than the initially slower PD axon. In contrast, PY may not explore this option, as the $Q_{10}$ of this slow axon is less than that of the faster PD.

Precise timing in the pyloric rhythm is essential to its function and the filtering behavior it controls [15, 23]. To test whether the pyloric axons maintain AP timing along their length, we calculated how much AP arrival times changed at each temperature. Fig 2B shows that AP arrival times were delayed by less than 20ms at the coldest temperature and arrived prematurely by less than 30ms at the warmest temperature. AP arrival times thus changed only moderately. When one considers the differences between axons, the changes in arrival times were even smaller (time difference between any given pair of neurons at a single temperature in Fig 2B). We next tested whether these modest changes affected the phase relationship between the neurons. Phase relationships provide a means to assess the structure of the rhythm independent of the frequency of the rhythm. In *Cancer borealis*, phase relationships are well-maintained across temperatures and between animals when measured centrally [14, 20, 24]. For example, the onset time of the individual neurons changes proportionately with cycle period, such that shorter cycle periods lead to shorter delays between an individual neurons' initiation, which maintains their relative timing [25]. We calculated the deviation in phase onsets using the changes in propagation velocities measured at different temperatures, with PD as the reference for the phase constancy line (Fig 2C). Consistent with the LP axon's slightly lower $Q_{10}$ (Fig 2A), LP's phase onset remained near constant throughout the temperature range tested (Friedman RM ANOVA of ranks, $X^2$ = 22.827, df = 8, N = 5, p<0.01, Dunnett's *post hoc* test, all comparisons ns). In contrast, PY's phase onset became progressively delayed as temperature increased, and significantly different from phase constancy above 19°C (one-way RM ANOVA, $F_{(8,32)}$ = 40.712, p<0.001, Dunnett's *post hoc* test, 10°C-19°C, 10°C-20°C, 10–23°C, and 10°C-25°C p<0.05, N = 5). The small changes in timing and phase between LP and PD and the larger change and sign of this change between PY and PD are thus consistent with the relative values of their respective $Q_{10}$s. However, despite statistical significance, PY's onset delay never exceeded the natural variability observed in ongoing rhythms (Fig 2C), suggesting that it was unlikely to interrupt the behavior. Thus, the changes to timing and phasing did not disrupt the physiological function of the pyloric rhythm in the temperature range tested.

## Mechanisms underlying temperature-robust AP propagation

The above results revealed surprisingly similar and modest temperature responses of the pyloric axons. Given that much higher temperature responses of axonal ion channels have been reported [12, 26] and that the pyloric neurons achieved low velocity $Q_{10}$s despite their distinct ion channel properties [27, 28], we asked what ion channel properties would support temperature-robust AP conduction velocities. First, we investigated how individual intrinsic properties affected AP propagation velocity when temperature changes. For this, we created a 3μm diameter, 4cm (50μm per compartment) model axon using Hodgkin-Huxley formalism [29], with voltage-gated Sodium and Potassium channels, as well as leak channels (see Materials and Methods). Temperature sensitivity was implemented by changing the maximum conductance of the channels (Sodium channel conductance: $\bar{g}_{Na}$, Potassium channel conductance: $\bar{g}_{K}$, or Leak channel conductance: $\bar{g}_{Leak}$) and channel gate time constants (Sodium activation gate time constant: $\tau_m$, Sodium inactivation gate time constant: $\tau_h$, Potassium channel activation

gate: $\tau_n$) relative to their baseline values at 10˚C. Our model was not aimed at a specific axon type but instead intended to generally address the question which intrinsic properties altered AP conduction velocity. A wide range of temperature responses of ion channel properties have been published, ranging from just above 1 to more than an order of magnitude higher in a selected few [26, 30]. We considered $Q_{10}$s of 1.5, 2, 3, or 4 for each channel property to represent increasingly more temperature sensitive properties and most axonal ion channels. We additionally considered that intrinsic membrane properties may change through extrinsic actions when temperature changes, including neuromodulatory influences [31] and slower influences such as increased ion channel expression [27]. While these influences are typically not reported in the form of $Q_{10}$ values, they can be well represented in our model through the same parameter changes because the model does not discriminate between fast and slow temperature influences on ion channels or between the mechanisms by which these influences act. For example, the $Q_{10}$s for the maximum conductance values of axonal ion channels are usually around 1.5 [9], but neuromodulatory influences can further augment maximum conductance through immediate and long-lasting actions, including channel transcription. In other words, the conductance $Q_{10}$ of individual channels remains the same but additional channels are recruited, effectively increasing the total conductance $Q_{10}$. We thus also considered values beyond the published $Q_{10}$.

At 10˚C the axon propagated APs at 1.28m/s. Fig 3A shows the minimum and maximum velocities resulting from changing the $Q_{10}$ of a given channel property at each temperature while maintaining all other values at canonical values. There were three sets of responses: (1) The Potassium channels had very little influence on velocity. (2) High temperature sensitivity of the maximum leak conductance ($\bar{g}_{leak}$) and the time constant of the Sodium inactivation gate ($\tau_h$) decreased propagation velocity. (3) High temperature sensitivity of the maximum Sodium conductance ($\bar{g}_{Na}$) and the time constant of the Sodium activation gate ($\tau_m$) increased propagation velocity.

Initially, we varied individual $Q_{10}$s of each ion channel property while maintaining all others at the baseline value. However, in biological axons these properties would all be affected by temperature at the same time, and not necessarily in the same way since even the activation and inactivation gates of a single channel can be distinct [32]. Due to the possibility of nonlinear interactions between these properties, we reasoned that when several properties change, the resulting velocity may not be directly obvious from the results of changing a single property. To address this issue, we varied all channel property $Q_{10}$s to capture all possible combinations in an exhaustive search (Fig 3B). However, because changes in the temperature sensitivity of the Potassium channel properties had essentially no influence on propagation velocity (Fig 3A), the maximum Potassium conductance and activation gate time constant remained at $Q_{10}$ = 1.5, consistent with previously reported values [33].

To discern the influence of individual ion channel properties on velocity, the data was sorted with increasing $Q_{10}$s for each channel property (Fig 3B, inset). Each colored square represents a model axon with a different combination of temperature-sensitive channel properties. Overall, Fig 3B shows similar results to those described when only a single channel property was temperature-sensitive (Fig 3A). Larger maximum Sodium conductance and activation gate time constant $Q_{10}$s caused larger increases in velocity, as depicted by a distinct change to warmer colors as the values of both increase along the horizontal axis at each temperature (e.g. a change from light green to red at 30˚C in Fig 3B). In contrast, changes in Sodium channel inactivation gate time constant and maximum leak conductance $Q_{10}$s decreased velocity, but only by a smaller amount, as shown by the modest change in color vertically (e.g. a change from light green to blue at 30˚C in Fig 3B).

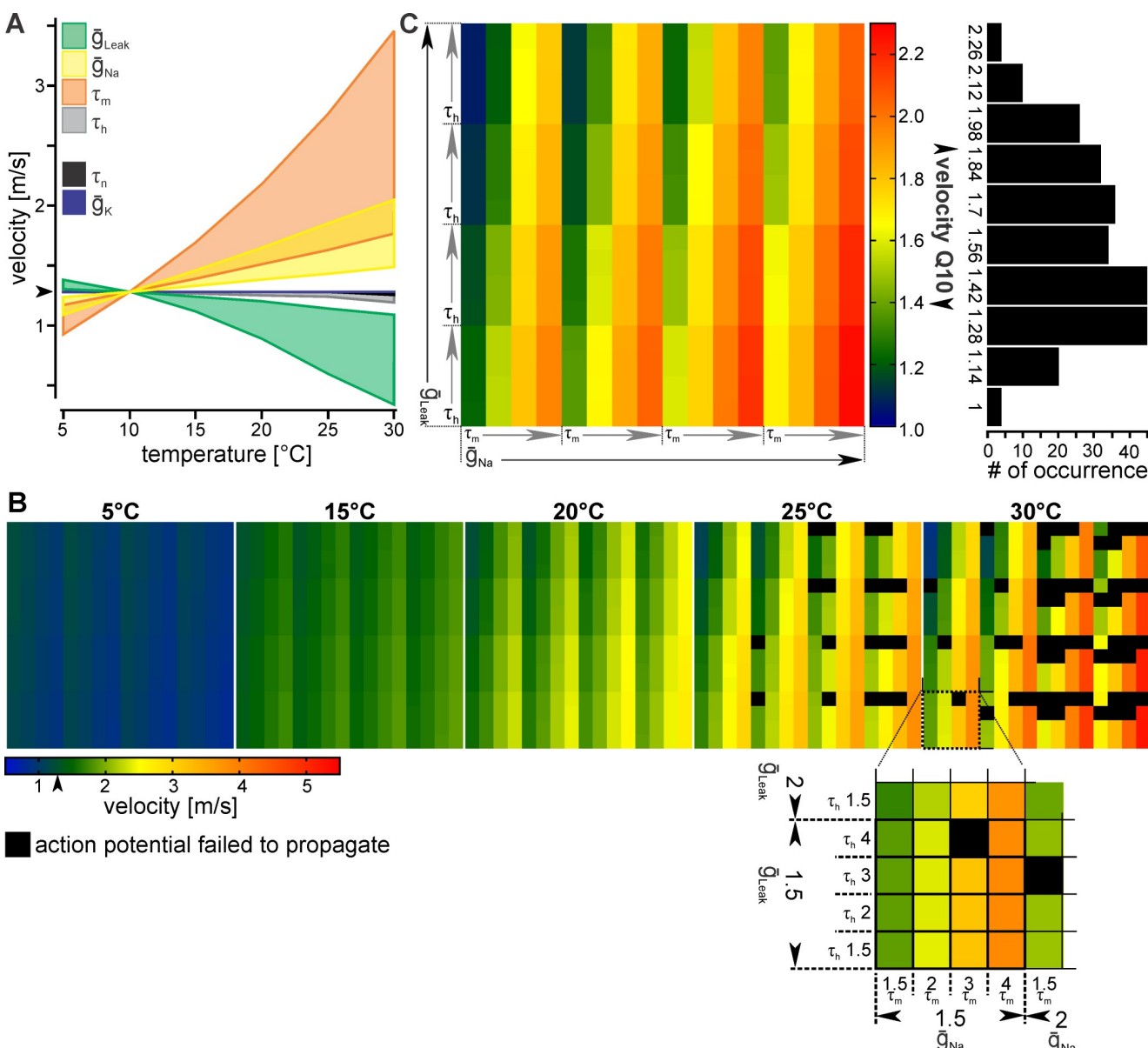

**Fig 3. Similar velocity changes in response to temperature can be achieved though ion channel properties with widely different temperature-sensitivities.** (A) Model prediction of how the temperature sensitivity of each channel property ($Q_{10}$ = 1.5, 2, 3, or 4) influences propagation velocity. Colored area shows the velocity range resulting at each temperature. The arrowhead identifies the propagation velocity of the axon at baseline temperature (10˚C). (B) Influence of different channel property $Q_{10}$s on propagation velocity. A single axon is shown at different temperatures. Each colored square represents an axon with a unique combination of ion channel property $Q_{10}$s. Colors give the measured AP conduction velocity. Sodium channel activation gate time constant and maximum conductance $Q_{10}$s increase horizontally, while maximum leak conductance and Sodium channel inactivation gate $Q_{10}$s increase vertically (bottom inset). (C) Left: Temperature-dependence of AP propagation velocity, measured as velocity $Q_{10}$s for each unique model. $Q_{10}$s were calculated from their velocities at 5 and 15˚C. Right: Quantification of total $Q_{10}$ occurrence.

In general, the velocities of most axons increased with temperature, though several failed to propagate APs at the higher temperatures (black squares in Fig 3B). The failing axons consistently had high Sodium inactivation gate time constant $Q_{10}$s. Considering that the temperature sensitivity of the Sodium inactivation gate time constant had little effect on velocity, we were surprised that it had such a strong influence on whether an AP propagates along the length of the axon. This indicates that while the speed at which the Sodium channel inactivates may not

strongly affect timing, it is nonetheless critical to temperature-robust axons, as it limits the temperature range at which APs can be propagated.

Surprisingly, we found a small minority of axons that slowed down at higher temperatures. These axons had higher $Q_{10}$s for their leak conductance than for their Sodium activation gate time constant. They showed consistently small velocity changes across most temperatures, with more pronounced drops in velocity at the highest temperature. They may represent axons pushed towards the upper limit of their functional temperature range, but have not yet failed to propagate action potentials [34], or represent conditions where leak increases exorbitantly and shunts spike initiation [35].

The fastest axons corresponded to high maximum Sodium conductance and activation gate time constant $Q_{10}$s (Fig 3C, bottom right), and the slowest to high maximum leak conductance $Q_{10}$s (top left). Importantly, in between these two extremes, many combinations resulted in similar propagation velocities, as indicated by large areas of similar color. The main conclusion from this analysis is that for a majority of the model axons, velocity increased only moderately. In fact, when we quantified the velocity $Q_{10}$ between 5 and 15 degrees the resulting range (1–2.3) was smaller than that of the individual channel properties (1.5–4). Most models fit into the mid-value range of 1.28 to 1.98 (on left: green to orange, on right greatest number of occurrences), suggesting that many combinations of ion channel property temperature sensitivities result in similar, modest velocity changes.

Similar to the biological neurons, we observed that the temperature dependence of some models did not seem to follow a single exponential for the entire temperature range but may be better described by two or more regimes (S3A Fig). To address this issue, we used the slope of five degree increments to describe the velocity temperature response accurately over the entire temperature range (S3B–S3F Fig). This quantification shows that while the minimum and maximum $Q_{10}$ values shifted at different temperatures, the total range of $Q_{10}$ values remained the same and the greatest occurring values were relatively unchanged. This further supported our conclusion that many combinations of temperature sensitivities resulted in modest increases in velocity and that the underlying ion channels are not restricted to a small range of $Q_{10}$s.

## Mechanisms that support temperature-robust timing between different diameter axons

When multiple neuron types are involved in the control of the same behavior, the temperature responses of all axons should contribute to how well AP timing between axons is maintained when temperature changes. For a single axon, many combinations of parameters resulted in low velocity $Q_{10}$s, and the Sodium channel activation time constant and its maximum conductance were the strongest influencers of axon velocity. To test whether there are similarly many combinations of ion channel properties and specific drivers that support temperature-robust AP timing between axons, we modeled two additional axons. These axons had identical ion channel properties and ranges of $Q_{10}$s, and thus also velocity $Q_{10}$s, but different baseline velocities. Different velocities were achieved by adapting axon diameter (6μm, 1.8m/s and 12μm, 2.1m/s). The range of diameters allowed us to match the velocity difference we observed in pyloric axons, but additionally enabled us to extrapolate results beyond the STNS, since the difference in velocity between interacting axons in other systems can even be greater. To determine how temperature-induced changes of velocity affected AP timing between axons, we first measured AP arrival times at a baseline temperature of 10˚C for each axon. We then determined the difference between arrival times of pairs of neurons, the 3μm and 6μm diameter axons (Fig 4A, top), and the 3μm and 12μm diameter axons (Fig 4A, bottom). We termed this

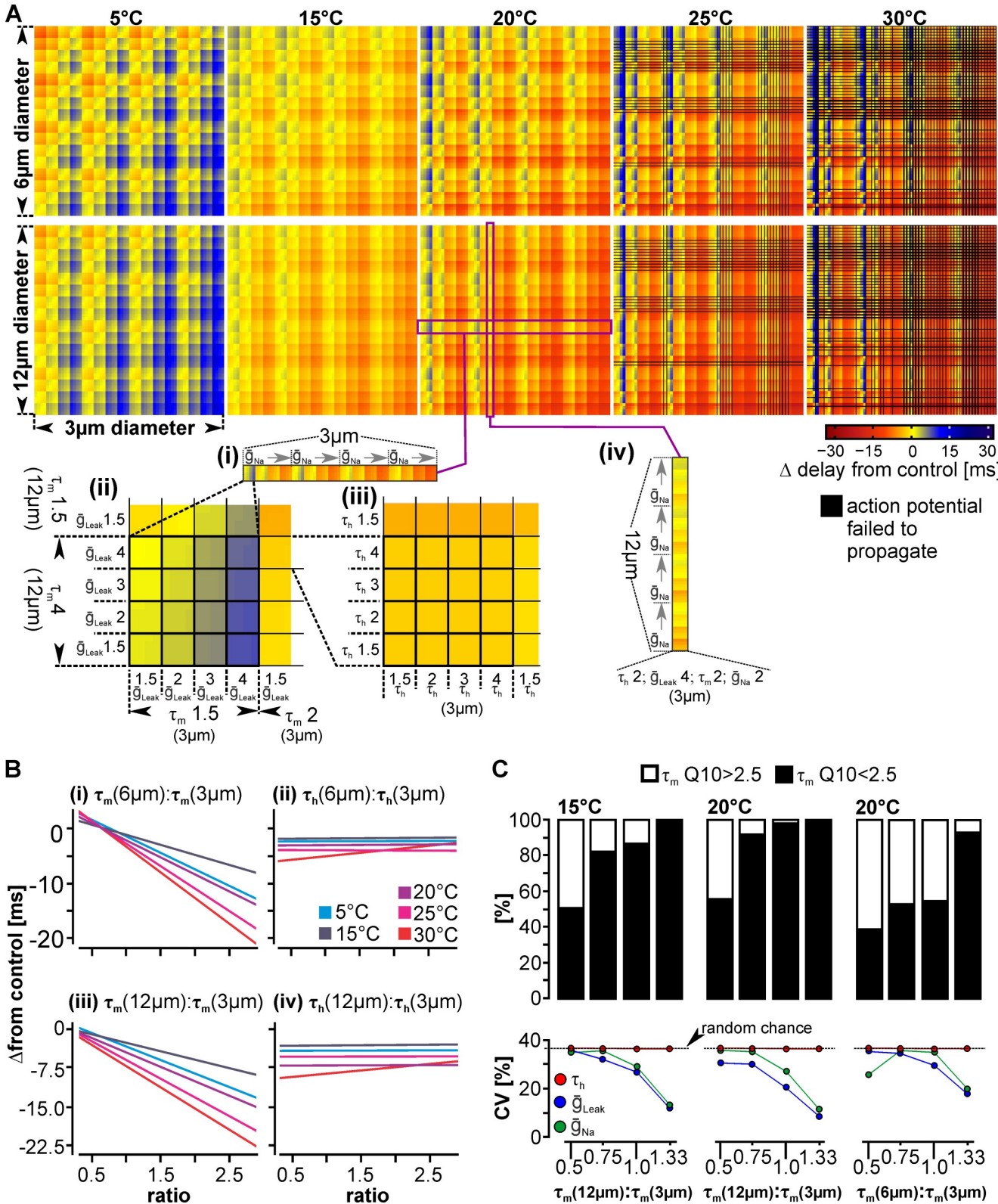

**Fig 4. Timing between axons depends on the temperature-sensitivity of channel properties in small and large diameter axons.** (A) Temperature-induced changes in AP timing between axons. The colors show how much AP arrival times change between axons at the axon terminal ('delay'). Delays were calculated in reference to the baseline temperature (10°C) for each axon and then as the difference (Δ) between two different diameter axons. Positive

delays (blue) denote APs on the slower, smaller diameter axon arriving late. Conversely, negative delays (red) denote APs on the smaller diameter axon arriving earlier. Yellow colors indicate no or only small changes to AP timing between axons. At each temperature, all channel property $Q_{10}$s of the 3μm axon are varied along the x-axis, with increasing $Q_{10}$s from left to right. On the y-axis, all channel property $Q_{10}$s of the 6μm (top) and 12μm (bottom) axon are varied, with increasing $Q_{10}$s. The bottom insets (i-iv) give the details of which properties were nested inside each other, and how they were varied. (i—iii) Example of 16 variations of ion channel property $Q_{10}$s in the 12μm axon (vertically, $g_{Na}$ = 2, $\tau_m$ = 4, $g_{leak}$ = 1.5–4, $\tau_h$, = 1.5–4). From left to right, all $Q_{10}$ combinations of the 3μm axon are shown (i), and then a subset of them (ii, iii). (iv) Example of how changes to the 12μm axon affect the delay with respect to a single 3μm axon ($\tau_h$, = 2, $g_{leak}$ = 4, $\tau_m$ = 2, $g_{Na}$ = 2). (Bi-iv) Delay as a function of the ratio between ion channel property $Q_{10}$s in both axons, representing the best (Sodium channel activation gate time constant, $\tau_m$) and the worst predictors of timing (Sodium channel inactivation gate time constant, $\tau_h$). Ratios were calculated by dividing the larger axon by the smaller. (C, top) The percentage of models that achieve temperature-robust timing with high Sodium channel activation gate time constant ($\tau_m$) $Q_{10}$s depends on ratio of $\tau_m$ between the two axons, the temperature, and difference in axon diameter. (C, bottom) When the Sodium channel activation gate time constant $Q_{10}$ is constrained to either high or low values, the coefficient of variation (CV) of the maximum leak and Sodium conductances (blue and green respectively) decreased from random chance (dotted line), indicating that coordinating these channel properties becomes more important for maintaining robust timing.

difference the 'delay' between AP arrival times. This delay changes with temperature and we plotted it in Fig 4A as color intensity and shade for each pair of compared axons, and for each temperature. Red indicates that APs in the smaller axon arrived prematurely, while blue indicates that they were late. Yellow represents that the delay did not change. Along the x-axis, we altered the $Q_{10}$s of the 3μm axon and we changed the $Q_{10}$s of the larger axons along the y-axis. Each axis has every possible combination of channel properties, with all $Q_{10}$ values for the Sodium inactivation gate time constant, leak conductance, Sodium activation gate time constant, and Sodium conductance. The insets in Fig 4A show the details of how we plotted these properties. From left to right, we first increased the $Q_{10}$ of Sodium conductance (1.5–4, Fig 4A inset i). Within each Sodium conductance $Q_{10}$, we increased Sodium activation gate time constant $Q_{10}$ from 1.5–4 (Fig 4A inset ii). Within each $Q_{10}$ of Sodium activation gate time constant, we increased leak conductance $Q_{10}$ from 1.5–4 (Fig 4A inset ii) and within each $Q_{10}$ of leak conductance, we increased Sodium inactivation gate time constant $Q_{10}$ from 1.5–4 (Fig 4A inset iii). For each temperature, all 256 combinations of $Q_{10}$s in the 3μm axon are thus plotted along the x-axis. Similarly, all 256 $Q_{10}$ combinations of the respective larger axon are plotted along the y-axis, resulting in a total of 65,536 comparisons at each temperature and for each pair of axons.

In general, there were fewer color changes vertically and more color changes horizontally, suggesting that timing between axons is more sensitive to changes in the properties of the smaller diameter axon. For instance, when looking at the example of Fig 4A (inset iv), changes in the channel properties of the 12μm axon (i.e. along the vertical axis) for a given 3μm axon caused the delay to change from -6.13ms to 2.53ms (a range of 8.66ms). This was much less than the development of delays when the properties of the 3μm axon was changed (along the horizontal axis). The delay in all 16 axons shown in insets (i)—(iii), for example, changed much more, with the smallest range being 17.33ms (from -9.43ms to 7.90ms). It is thus mostly the properties of the smaller axon that determine whether temperature-robust timing can be achieved. Nevertheless, the fact that there were color changes due to the temperature sensitivities of channel properties in both the smaller and larger axon indicates that the combination of parameters and subsequent changes in propagation velocity in both axons, rather only one axon, determine how well timing is maintained.

We also observed that at each temperature, there were fewer yellow regions and more red or blue regions of stronger intensity colors in the 3μm to 12μm comparison (bottom) than the 3μm to 6μm comparison (top). This indicates that with larger initial velocity differences, fewer axons were able to maintain the expected AP delay. A similar effect was seen when temperature increased: in both comparisons the amount of yellow decreased (compare yellow regions in 15˚C to 30˚C), indicating that the further the temperature deviated from baseline, the fewer axon pairs that maintained timing.

## What channel properties need to be coordinated between different axons to support temperature-robust timing?

Our analysis of the individual axon showed that the channel properties do not have equal influence on propagation velocity, and the differential weighting of channel properties ultimately determines the propagation velocity for a given axon and temperature (Fig 3). Therefore, we deduced that the precision of timing between two axons may be determined primarily by the temperature sensitivity of the strongest predictors of propagation velocity, rather than the entire set of parameters in both axons. We further hypothesized that coordination between the strongest predictors rather than channel properties in a single axon would determine the precision of timing between the axons. By this token, we anticipated that the ratio of the Sodium activation gate time constant between the small and large diameter axon best predicts the delay. We thus plotted delay as a function of this ratio and fitted a linear regression. We used the slope of the regression to assess how well the ratio predicted delay, and then compared the results to those of other ratios, with steeper slopes indicating a better prediction. We analyzed all possible ratios of ion channel property $Q_{10}$s (S1 Table). Fig 4B compares the result for the best and worst predictors: the ratios of the Sodium activation gate time constant $Q_{10}$s (Fig 4Bi, iii) and the Sodium inactivation gate time constants $Q_{10}$s (Fig 4Bii, iv). For all temperatures, the activation gate time constant had the steepest slope and thus the strongest influence on timing while the inactivation gate time constant had the weakest. The ratios of other channel properties also produced significant linear regressions, but their slopes were less steep (S1 Table). This suggests two things: The temperature responses of the Sodium activation gate had the greatest influence on timing between the axons. Two, even though weaker, other ratios may either by themselves or in combination with others alter how well timing is maintained between axons. Of note was the ratio between the temperature sensitivities of the Sodium activation time constant of the larger axon and the maximum Sodium conductance of the smaller axon, which predicted the delay nearly equally well (S1 Table).

We thus asked whether coordinating the strongest predictors alone is necessary and sufficient to achieve temperature-robust timing. We defined temperature-robust as less than 5% deviation from control. At 20˚C this corresponded to a maximum deviation of 1.3ms. Consistent with Fig 4A, there were more temperature-robust models between the 3μm and 6μm axons than between the 3μm and 12μm axons. This was the case across temperatures as well, though the total number of temperature-robust models diminished with higher temperatures. For example, there were 10,293 (of 65,536 total) temperature-robust models at 20˚C between the 3μm and 6μm axons and only 7,344 between the 3μm and 12μm axons. At 30˚C, only 3,646 models (3μm and 6μm) and 2,281 models (3μm and 12μm) remained. Together this emphasizes the need for better coordination between more disparate diameter axons, and wider temperature ranges.

If ratios were not important, they should be represented equally in the pool of temperature-robust models. The minimum possible ratio in the entire database was 0.375 ($Q_{10}$ = 1.5 in the larger axon, and $Q_{10}$ = 4 in the smaller) and the maximum was 2.67 ($Q_{10}$ = 4 in the larger axon, and $Q_{10}$ = 1.5 in the smaller). For example, at 20˚C, we found two channel properties with constrained ratio representation, suggesting that the coordination of these was more likely to determine temperature-robust timing. The ratio between the Sodium activation time constants was limited to 0.375 through 1.5 (3μm vs. 6μm) and 0.375 through 1.33 (3μm vs. 12μm). Likewise, the ratio between the Sodium activation gate time constant and the maximum Sodium conductance was constrained to 0.375 through 2 (3μm vs. 6μm) and 0.375 through 1.33 (3μm and 12μm).

We tested whether coordinating either of these two channel property combinations was necessary to achieve temperature-robust timing. If necessary, the two axons require their ratio to be within a particular range and axons without the observed ratios will not yield

temperature-robust timing. We first tested this for the Sodium activation gate time constant ratios (ratios between 0.375–1.33). At 20˚C we found that 99.59% (3μm vs. 6μm axons) and 100% of (3μm vs. 12μm axons) of the models without these ratios were outside of the temperature-robust timing range. Similarly, when we tested the necessity of the Sodium activation gate time constant and maximum conductance ratio (0.375–1.33), at 20˚C we found that 96.92% (3μm vs. 6μm axons) and 100% (3μm vs. 12μm axons) of the models without these ratios were outside of the temperature-robust timing range. This suggests that both ratios were necessary to achieve temperature-robust timing, at least for the 3μm vs. 12μm axon comparison. If this was true, then _both_ ratios should be constrained in temperature-robust models. Indeed, we found that this was the case (3μm vs. 12μm axons: 100% = 7,344 of 7,344 temperature-robust models; 3μm vs. 6μm axons: 94.5% = 9,722 of 10,293 temperature-robust models).

We additionally tested whether coordinating the temperature sensitivities of either of these two channel property combinations was sufficient to achieve temperature-robust timing. In this case, all axons with the observed ratios will yield temperature-robust timing, independently of the other properties. We first tested the sufficiency of the Sodium activation gate time constant ratios. At 20˚C only 20.80% of the 3μm vs. 6μm axons and 14.94% of the 3μm vs. 12μm axons with these ratios resulted in temperature-robust timing. We found a similar result when we tested the sufficiency of the ratios between Sodium activation gate time constant and maximum conductance. At 20˚C, only 19.92% of the 3μm vs. 6μm axons and 14.94% of the 3μm vs. 12μm axons with these ratios resulted in temperature-robust timing. This indicates that neither of these combinations alone was sufficient to yield temperature-robust timing. As a follow up, we thus tested the sufficiency when both combinations were present together. However, percentages only mildly improved (23.74% of the 3μm vs. 6μm axons and 17.93% of the 3μm vs. 12μm), indicating that the temperature responses of other channel properties need to be additionally coordinated to ensure that the axons maintain temperature-robust timing.

This leads to two interesting conclusions: (1) although not constrained to a subset of combinations, the other channel property $Q_{10}$s matter to timing. (2) Despite the Sodium activation gate time constant causing the greatest velocity increase (Fig 3A), temperature-robust models can be produced with ratios of larger than one, i.e. where the Sodium activation gate of the larger and faster axon is more temperature-sensitive than that in the slower axon. This is surprising because it seems contrary to the idea that timing is maintained between different diameter axons when the larger is less temperature-sensitive. Together these observations lead to two predictions for ratios larger than one: (1) the faster axon's Sodium activation gate time constant $Q_{10}$ must be limited to small values to minimize the increase in velocity and (2) the other channel properties counterbalance an exorbitant velocity increase in the larger axon.

To test the first prediction, we evaluated the percentage of high (3, 4) vs. low (1.5, 2) $Q_{10}$ values of the Sodium activation gate time constant at ratios which could be achieved by more than one combination (for example, a ratio of 0.75 can be achieved by $Q_{10}$s of 3 (larger diameter axon): 4 (smaller diameter axon) or 1.5: 2). Fig 4C shows that as the ratio increases there is a shift toward lower $Q_{10}$s in both axons. Specifically, for the 12μm and 3μm axons at 15˚C and a ratio at 1.33, only $Q_{10}$s of 2: 1.5 yielded temperature-robust timing, whereas $Q_{10}$s with the same ratio, but higher individual values (4: 3) did not. In contrast, at a ratio of 0.5, values of both 2: 4 and 1.5: 3 were represented almost equally (Fig 4C, left top). At 20˚C, $Q_{10}$s were even more constrained and almost all axon pairs with ratios of 0.75 and 1.0 had small $Q_{10}$s. This demonstrates that larger temperature changes require more restricted $Q_{10}$s (Fig 4C, middle top). When we compared the 6μm axon to the 3μm axon at 20˚C, we found similar, but weaker trends, indicating that $Q_{10}$s are less restricted when axon diameters differ less (Fig 4C right top). Thus, as the larger axon becomes more temperature-sensitive compared to the smaller axon (ratios larger than one), it becomes restricted to lower $Q_{10}$s.

We also found evidence for a shift to higher $Q_{10}$s when the smaller axon was more temperature-sensitive than the large one (at a ratio of 0.5). In the 3μm to 6μm axon comparison, 61.13% of the models observed had high $Q_{10}$s. $Q_{10}$s were thus no longer free to vary randomly and high $Q_{10}$s were favored in the larger axon to match the increasingly faster processes in the smaller axon.

As an indicator for how influential the other channel properties were, we measured the coefficient of variation for each channel property $Q_{10}$ and compared them for different ratios of the Sodium activation gate time constant. Channel properties with no influence should occur randomly, while those that facilitate good timing should be restricted to certain values and thus vary less. We found that the coefficient of variation for the Sodium inactivation gate time constant always occurred randomly and was thus not was predictive of timing between the axons at any of these ratios (red line in Fig 4C, bottom). In contrast, the coefficient of variation for the maximum Sodium and leak conductances of the larger axon decreased from random chance (Fig 4C, bottom), and only did so at ratios where the Sodium activation gate time constant $Q_{10}$s were confined to either high or low values (when the black bars in Fig 4C (top) were either markedly above or below 50%). This suggests that the temperature sensitivities of these two properties become more influential to maintaining AP timing at these ratios, since they are no longer free to vary at random.

In conclusion, timing between different diameter axons at different temperatures is most affected by the coordination of channel properties whose temperature sensitivities have the largest impact on propagation velocity: the Sodium activation time constant and Sodium conductance. Coordinating the Sodium activation gate time constant of the larger axon with either Sodium activation gate time constant or the maximum Sodium conductance of the smaller axon is necessary for temperature robust timing between the axons. However, coordinating these properties alone is not sufficient. When the Sodium activation gate time constant of the larger axon is more temperature sensitive than the Sodium channel properties of the smaller one, the maximum Sodium and leak conductances of the larger axon become more influential on the precision of timing between the axons. These additional channels prevent an exorbitant increase the larger axon's propagation velocity, and only then results in temperature-robust timing between the axons.

## Discussion

### How can axons remain functional and support time-critical processes when temperature changes?

We investigated the effects of temperature on the axons of three stomatogastric motor neuron types with unique AP propagation velocities. Each neuron type is essential for controlling the filtering of food through the crab's pylorus [36]. Their timing is critical to the animal's survival as the PD, LP, and PY phase relationship is consistent across time [37], animals [15, 38], and temperature [14, 20], and even small changes in AP timing alters pyloric muscle activity [39]. This suggests that their relative timing not only requires the appropriate timing of spike initiation but also must account for propagation velocity differences between the axons as they project to their target muscles. Indeed, we found that pyloric axons are almost temperature insensitive ($Q_{10}$s ~1.5, Fig 2A) and that the phase relationships remained unchanged throughout the entire temperature range the animals experience (Fig 2C).

Axons face two general problems when temperature changes: Individually, each neuron type needs to reliably propagate APs to the periphery without failures. Additionally, AP timing of a neuron's own activity and the timing between neurons must be maintained to enable time-sensitive postsynaptic processes. The problems posed by temperature changes in axons

can at least partly be attributed to membrane excitability. Membrane excitability is a major contributor to spike timing as it affects summation and local spread of synaptic signals. This problem may be relatively smaller in axons as AP propagation does not depend on summation but rather on (all-or-nothing) depolarization of the previous axon site [40]. Loss of excitability, for example, via early Sodium channel inactivation or an increase in leak current, may not play a major role for velocity, but instead affect whether propagation occurs. Our model indeed predicts that there is a hierarchy to the influence that channel properties and their temperature responses have on propagation velocity (Fig 3A). High Sodium inactivation gate $Q_{10}$s led to AP failures along the axon, signifying that an increasingly early inactivation at high temperatures limited Sodium current and thus also the AP amplitude. This, in turn, led to insufficient depolarization further along the axon and a failure of AP initiation. This was particularly the case when accompanied by high leak $Q_{10}$s that further shunted AP amplitude. In contrast, how quickly the Sodium channels activate appeared more important for propagation since velocity depended on how quickly the voltage reaches its peak and how quickly thereafter a new AP can be generated. Indeed, the activation gate time constant and the maximum Sodium conductance were the best predictors of propagation velocity and critical for temperature-dependent velocity changes.

## A large range of ion channel $Q_{10}$s support low axon temperature-sensitivity

One of the most striking predictions of our model is that many different parameter combinations result in low propagation velocity $Q_{10}$s. This is surprising since we varied channel property $Q_{10}$s up to a factor of 4, covering the typical range of voltage-gated ion channels (with the exception of channels involved in temperature sensing, such as TRP and related channels [11, 41]). Making the temperature sensitivity of all the channel properties independent of one another allowed us to separate the influence of each component on velocity. Independent temperature sensitivities of different gates in the same channel have been described for several channels, including voltage-gated Sodium, Potassium and Calcium channels [33, 42]. For example, the activation and inactivation gate time constants of the voltage-gated Sodium channels expressed at the node of Ranvier in *Xenopus laevis* have $Q_{10}$s of 2.34 and 2.90 respectively [43].

Our results are reminiscent of recent findings that ion channel expression in a given neuron can differ substantially between animals and still produce similar activities [17, 44–46]. It appears that instead of a single optimum set of ion channels, similar activities are generated by correlated channel expression and conductance levels [27, 47, 48]. For example, the expression of the Sodium channel protein *para* varies more than 5-fold in stomatogastric motor neurons [18] and the expression of the two Potassium channel proteins *shab* and *shaw* by even more. Based on immunostainings this variability appears to extend to the axons [49–51]. Despite evidence that LP, PD and PY may vary substantially in their intrinsic properties [18, 45], we report low velocity $Q_{10}$s for all three neuronal types. Likewise, our models with distinct channel properties shows an abundance of similar velocity $Q_{10}$s (Fig 3A) indicating that a neuron may assume certain properties in one individual, but a different set in the next without changing the overall temperature response.

## Coordination of channel $Q_{10}$s enables temperature-robust timing between axons

In unmyelinated axons, diameter plays a major role in establishing the initial velocity differences between the neuron types. The model corroborates this as changes in ion channel conductance levels caused more minute changes in velocity, compared to differences in axon

diameter. However, how temperature alters the timing between axons depends on how the $Q_{10}$s of the different axons compare. The pyloric central pattern generator creates a stable and consistently patterned activity over a fairly broad temperature range, with a constant phase relationship of the PD, LP and PY neurons [14, 24]. This trait is achieved through coordinated $Q_{10}$s of each neuron's intrinsic properties [14, 20]. For example, the $Q_{10}$s of $I_A$ and $I_h$, two ionic conductance with antagonistic function, are co-regulated, such that temperature-induced increases in one are counterbalanced by a similar increase in the other [14]. Modeling studies further indicate that the specific $Q_{10}$ values associated with the ion channels are relatively unimportant, as long as they are properly coordinated [21]. Reminiscent of these data, our model shows that velocity $Q_{10}$s of slower axons must be higher than in faster axons to maintain proper timing. This constraint, however, does not necessarily require channel property $Q_{10}$s to be constrained. While our model predicts that coordinating the Sodium conductance and activation gate time constant $Q_{10}$s between the involved neurons is necessary for temperature-robust timing, it is not sufficient. Other channel properties also impact how precisely timing is maintained and are critical to preventing AP failures. Generally, ratios where the Sodium conductance and activation gate time constant $Q_{10}$s are smaller in faster axons than in slower axons (ratios of <1) favor the maintenance of timing between neurons (Fig 4C). However, in select cases, the temperature-responses of other channel properties can determine AP timing between the axons. For example, at ratios >1, proper timing can only be achieved if both, the Sodium conductance and activation gate time constant $Q_{10}$s are close to 1. The near temperature-insensitivity in the faster axon prevents exorbitant velocity increases and allow slower axons to keep up. In this scenario, the leak conductance gains a comparatively stronger influence on AP timing (Fig 3C, bottom), although this only occurs when its $Q_{10}$s are high. This scenario is tempered by the fact that strong leak currents cause shunts and propagation failures (Fig 2B; [52, 53]), unless accompanied by strong depolarizations as well. Consequently, our model predicts that the more disparate two axons are in initial velocity, the fewer velocity $Q_{10}$ combinations are available that support temperature-robust timing between them, especially across large temperature ranges.

In the context of transient temperature changes coordinating ion channel properties to maintain timing is important for both myelinated and non-myelinated axons. While membrane capacitance has a larger influence on velocity than channel properties in myelinated axons, temperature-induced changes to capacitance are negligible. Nevertheless, the propagation velocity of myelinated axons changes with temperature, suggesting that other temperature sensitive processes, including the channel gates, are responsible for that change. Quicker activation at nodes of Ranvier results in faster AP propagation velocities [54] indicating that the determinants of temperature-dependent velocity changes in myelinated and non-myelinated axons are similar. As such, the pyloric axons and their natural dependence on ambient temperature are an attractive test bed for temperature effects despite their low propagation velocities. While the low speed may seem to disqualify them from functional interpretations of several fold faster vertebrate axons, there is ample evidence that proportionally small changes in timing play significant roles in fast vertebrate processes. This is best exemplified by recent findings that the axon initial segment and myelination change in activity-dependent ways [55].

An alternative to achieve coordinated temperature responses of ion channels could be axon-specific neuromodulation. Neuromodulators act on ionic conductances in all neuronal compartments, including axons, on various time scales, from long-term homeostatic changes to rapid transient ones [56]. Neuromodulators have been demonstrated to differentially affect axons in the same circuits, where they alter membrane excitability and propagation velocity [6, 49]. With respect to reliable propagation and AP failures, maintaining local excitability is imperative at high temperatures, and we have shown that peptide modulators in the STG can

maintain neuronal excitability during temperature-induced increases of the leak current [35]. Modulating axonal ion channels can also directly alter AP propagation velocity [31]. With a temperature-dependent release, modulation could provide a mechanism to change the temperature response of the axon and thus, while not immediately obvious, alter its effective $Q_{10}$. For example, the Sodium conductance $Q_{10}$ could be increased by a temperature-dependent activation of Sodium channel transcription or de-novo expression, thus altering its maximum conductance with temperature. Furthermore, axons can express different variants of Sodium channels [57, 58], and it is at least conceivable that a differential expression of channels with distinct $Q_{10}$s or the additional expression of other Sodium channel variants such as a persistent Sodium channel could lead to changes in the Sodium conductance [59] and thus the axon's temperature response. Independent of the underlying mechanisms, coordinating even temperature-sensitive Sodium channel properties supports temperature-robust timing between axons.

## Materials and methods

### Dissection

Adult male crabs (*Cancer borealis*) were purchased from The Fresh Lobster Company (Gloucester, MA) and kept in tanks with artificial sea water (salt content ~1.025g/cm$^3$, Instant Ocean Sea Salt Mix, Blacksburg, VA) at 11˚C and a 12-hour light-dark cycle. Before dissection, animals were anesthetized on ice for 20–40 minutes. The stomatogastric nervous system (Fig 1A) was isolated from the animal following standard procedures, pinned out in a silicone lined (Sylgard 184, Dow Corning) petri dish and continuously superfused (7–12 ml/min) with physiological saline (10˚C). In some experiments the dorsal ventricular nerve (*dvn*) was transected (Fig 1A). A part of the lateral ventricular nerve (*lvn*) was bathed in physiological saline at temperatures between 5˚C and 25˚C inside of a petroleum jelly well at least three times for each temperature. *C. borealis* saline was composed of (in mM) 440 NaCl, 26 MgCl2, 13 CaCl2, 11 KCl, 11.2 Trisma base, 5 Maleic acid, pH 7.4–7.6 (Sigma Aldrich).

### Extracellular and intracellular recordings

Petroleum jelly-wells were built to electrically isolate a small part of the nerve from the surrounding bath [60]. One of two stainless steel wires was placed inside the well to record neuronal activity of all axons projecting through that nerve. The other wire was placed in the bath as reference electrode. Extracellular signals were recorded, filtered, and amplified through AM Systems amplifier (Model 1700, Carlsborg, WA). Files were recorded, saved, and analyzed using Spike2 Software (CED, UK; version 7.18) at 10 kHz. The activity of the pyloric neurons was monitored on multiple extracellular recordings simultaneously. Specifically, we recorded APs from the *dvn* near the location of confocal imaging, on the lateral ventral nerve (*lvn*) just after the axons bifurcate into two bilateral branches approximately 1.5cm from the STG, and on the *lvn* just before the axons project into separate nerves near their respective peripheral muscles (Fig 1A).

Extracellular amplitudes were measured as the voltage difference between trough and peak of the extracellular waveform. Extracellular waveform shape and amplitudes depend on the properties of the recording well and thus differ between recording sites and animals. Thus, to compare amplitude ratios of the pyloric neurons between different recording sites and between animals, amplitudes were normalized to the average amplitude of the three pyloric neurons at a given location.

Baseline AP propagation velocity was calculated at a constant temperature (10˚C) by stimulating each axon individually and recording the elicited APs at two locations along the axon

(see Fig 1A). For experiments where temperature effects on velocity were measured, the stimulation temperature was kept constant at 10˚C while the rest of the nerve between the two recording sites was bathed in temperature-controlled saline separated by an additional well. Velocity changes were measured continuously, with temperature ramps ranging between 0.01˚C/s and 0.3˚C/s.

$Q_{10}$ values were used to compare changes in propagation velocity across temperatures. The $Q_{10}$ is the conventional measure to determine the temperature sensitivity of chemical and biological processes and is defined as how much a given rate changes with a temperature change of 10˚C. To calculate the propagation velocity $Q_{10}$, we plotted velocity against temperature in a semilog plot. The $Q_{10}$ was then extracted from the slope of the linear regression using the equation: $Q_{10} = 10^{10*slope}$ [14].

Changes in phase relationships of the pyloric neurons were calculated using their phase onsets only. Measured conduction velocities of the three pyloric neurons at the different temperatures were converted into times at which APs arrive at their peripheral muscles using the approximate distance between somata in the STG and the peripheral muscles (4cm). The time differences between LP and PD, and PY and PD, were calculated and converted into phase using published pyloric cycle periods at each temperature [14]. Since PD is part of the pacemaker ensemble of the pyloric rhythm [25], we used it as our reference for calculating the phase constancy line and deviations from it. Phase constancy was calculated relative to the cycle periods at each temperature. The natural variation in phase during ongoing pyloric rhythms was used to calculate the size of the standard deviation at each temperature. For this, the phase onsets of at least 10 cycles of unperturbed pyloric activity were averaged (N = 6 animals).

To facilitate intracellular recordings and dye injections, we desheathed the stomatogastric ganglion (STG) and visualized STG somata with white light transmitted through a dark field condenser (Nikon, Tokyo, Japan). Intracellular recordings were obtained from cell bodies using 20–30 MΩ glass microelectrodes (Sutter 1000 puller, 0.6M $K_2SO_4$ + 20mM KCl solution) and an Axoclamp 900A amplifier (Molecular Devices) in current clamp mode. For dye injections, electrodes tips were filled with 10mM Alexa Fluor 568 (PD, red), Alexa Fluor 488 (LP, green), or a mix of the two (PY, yellow, Life Technologies, Grand Island, NY) in 200mM KCl. Repetitive negative current pulses ranging between -3 and -5nA with pulse durations of 1–2 seconds for 30 mins were used to iontophoretically inject the dyes. Individual neurons were identified by comparing AP occurrence between intracellular and extracellular recordings, and by their known membrane voltage waveforms [25, 36].

## Extracellular axon stimulation

We used retrograde extracellular nerve stimulation to elicit APs in pyloric axons as described in detail by Städele and colleagues [61]. Current pulses were applied with a Master-8 pulse stimulator (A.M.P.I., Israel) at a frequency of 1Hz to avoid history-dependent effects on conduction velocity. To determine changes in conduction velocity, we averaged at least 5 consecutive stimuli at the same temperature. Stimulus parameters were 1 ms for pulse duration and 0.2 to 1 V for stimulus amplitudes. Specificity of stimulation was achieved by selectively stimulating nerves that contain only a single axon type. For PD, the *pdn* was stimulated, for PY, the *pyn* was stimulated. Specificity was confirmed by assessing the presence of APs on the respective contralateral nerves. For LP, the posterior section of the *lvn* ($lvn_{low}$) was stimulated at the lowest threshold that elicited APs. LP is the largest axon in this section of the nerve, with the lowest stimulation threshold. Selective stimulation was confirmed by the presence of a large amplitude AP on the contralateral *lvn*, and the absence of (additionally stimulated) APs on the *pdn* and *pyn*.

## Confocal imaging

The tissue was scanned using a Zeiss SP8 laser scanning confocal microscope using an oil objective (HC PL APO 40x/NA 1.30 Oil CS2). Images were obtained using 3.7Mpixel field of view, z-resolution 250 nm. ImageJ (FiJi) [62] software was used for image processing and maximum intensity projections, 3D analysis and diameter measurements. We measured the average diameter of each axon in the visible region by calculating the area of the axon and dividing by the length of the axon. In addition, axon diameters were measured from non-confocal imaging data. In some animals, axons showed a beaded structure. In this case, we measured in-between beads.

## Computer model

To determine what ion channel properties could support temperature robust timing, we created a model axon with Neuron [63]. The model is available on ModelDB (accession number 260972). The total axon length was 4.075cm (50μm/segment), and had an axial resistance of $28\Omega^*$cm. Current pulses stimulated single APs in a temperature-insensitive axon region to ensure that observed changes in propagation were not due to changes in AP initiation. APs then propagated into a temperature-modified region of the axon and arrival time was measured at two locations: 0.5cm from the temperature-insensitive region and 1.5cm from the axon end. This prevented influences from the stimulation current and from artifacts due to the end of the model axon. Changes in timing were considered for an axon length of 4cm for all axons, which is the approximate distance APs travel on the pyloric axons before they reach their synaptic terminals.

Passive neuron parameters and maximum ion channel conductances at baseline temperature were the same throughout all regions of the axon; membrane capacitance: 1 μF/cm$^2$, resting membrane potential: -60 mV, leak conductance ($\bar{g}_{leak}$): 0.0016 S/cm$^2$, Sodium conductance ($\bar{g}_{Na}$): 0.48 S/cm$^2$, and Potassium conductance ($\bar{g}_K$): 1.088 S/cm$^2$. Active membrane properties were implemented according to Hodgkin-Huxley equations and modified to include a temperature sensitivity factor for specific ion channel parameters in the temperature-sensitive region. Activation/inactivation were implemented using $inf_{ion} = 1/(1+\exp(-k(V_{1/2}-V)))$ and time constants were implemented using $\tau_{ion} = A\times\exp(k(V-V_{1/2}))$ with V being the membrane voltage, $V_{1/2}$ being the half maximum potential, k being step size, and A being a scaling factor for the time constant. Temperature sensitivity was implemented for the maximum ion channel conductances by: $\bar{g}_{ion}(T, Q_{10}) = \bar{g}_{ion} \times R$, and in the ion channel time constants by: $\tau_{ion}(T, Q_{10}) = \frac{1}{R}(\tau_{ion})$, where $R$ is a factor of temperature sensitivity at a single temperature. The temperature sensitivity factor was determined using the equation: $R = 10^{\frac{\log(Q10)}{10/(T-10)}}$, where $Q_{10}$ is the temperature sensitivity and had values of 1.5, 2, 3, or 4, and T is the temperature of the axon and had values of 5, 15, 20, 25, or 30˚C. Values for the Sodium channel activation were $k_{inf}$: -0.4 /mV, $V_{1/2\ \bar{g}}$: -36mV, $A_\tau$: 2 ms, $k_\tau$: -0.5 /mV, $V_{1/2\ \tau}$:-40 mV. Values for the Sodium channel inactivation were $k_{inf}$: 1 /mV, $V_{1/2\ \bar{g}}$: -39.5mV, $A_\tau$: 40 ms, $k_\tau$: -0.025 /mV, $V_{1/2\ \tau}$: -55 mV. Values for Potassium channel activation were $k_{inf}$: -0.125 /mV, $V_{1/2\ \bar{g}}$: -33 mV, $A_\tau$: 55 ms, $k_\tau$: -0.015 /mV, $V_{1/2\ \tau}$: -28 mV.

## Data analysis and statistical tests

Data were analyzed using scripts for Spike2 (available on www.neurobiologie.de/spike2) and by using built-in software functions. Statistical tests were performed using SigmaStat (version 12, Systat Software GmbH, Erkrath, Germany). Kolmogorov-Smirnov test with Lillifors correction was used to assess normal distribution of data sets. One-way repeated measure

ANOVA with Holm-Sidak posthoc or Student-Newman-Keuls *post hoc* test were used to test for significant differences. Friedman repeat measure ANOVA of ranks with Dunnett *post hoc* tests were used to test for significant differences in data without normal distributions. Statistical results are reported in the format: statistical test, F(degrees of freedom, residual) = f value, p value, posthoc test, number of experiments. "N" denotes the number of preparations, while "n" is the number of trials/APs. Significant differences are indicated using $^*$p<0.05, $^{**}$p<0.01, $^{***}$p<0.001. Exact p values are given unless they were smaller than 0.001, in which case p<0.001 is indicated. Post-hoc tests were carried out for a significance level of p<0.05 unless otherwise stated. Type of experimental design: Random. Data were plotted either with Excel (Microsoft) or Gnuplot (gnuplot.info), and edited with Coreldraw (version X7, Corel Corporation, Ottawa, ON, Canada).

## Supporting information

**S1 Fig. Comparison of action potential propagation velocity in axons with different diameters but similar temperature sensitivities.** At velocity $Q_{10}$s greater than one, the difference in velocity between neurons changes with temperature even when all axons have the same $Q_{10}$. (A) With $Q_{10}$s of one, the velocities of three different diameter axons are equally insensitive to temperature changes (left, middle). Thus, they show identical velocity differences at all temperatures (right, black bars). Colors denote different axon diameters. (B) At $Q_{10}$s larger than one, the velocities of the three axons still change proportionately (left, middle), but the difference in velocity between neurons increases (right, blue bars).
(TIF)

**S2 Fig. Sigmoidal curve fits the temperature response of the pyloric axons over the entire temperature range the crab experiences.** Velocity is plotted and sigmoidal fits for the pyloric axons are given by the equations, PD (red): $velocity_{PD} = -0.260 + \frac{0.417}{1+e^{\frac{-(Temperature-13.172)}{4.186}}}$, PY (yellow): $velocity_{PY} = -0.355 + \frac{0.370}{1+e^{\frac{-(Temperature-12.519)}{3.963}}}$, and LP (green): $velocity_{LP} = -0.189 + \frac{0.401}{1+e^{\frac{-(Temperature-12.561)}{4.371}}}$.
(TIF)

**S3 Fig. The velocity Q10 range remains consistently small independent of where the rate of change is measured.** (A) The velocities of two example neurons are plotted on a logarithmic scale to show when a single $Q_{10}$ value fits well (i) and when it does not fit well (ii). $Q_{10}$ values are calculated based on the linear regression of the entire temperature range. (B-F) The distribution of velocity $Q_{10}$s measured in five degree Celsius increments shows a small range in $Q_{10}$ values with a majority lying within a range of 1.2–1.8.
(TIF)

**S1 Video. Simultaneous confocal imaging of three pyloric axons.** Axons were imaged in the *dvn* after dye injection into their somata. LP: green, PD: red, PY: yellow.
(MP4)

**S1 Table. The slope and $R^2$ of the linear regression associated with all possible combinations of ion channel property $Q_{10}$ ratios.** A linear regression was formed for every possible ratio of large to small axon channel properties at each temperature. All ratios are shown as the larger axon property over the smaller axon property.
(TIF)

**S1 Data. Raw data for all figures are included in this Excel sheet.**
(XLSX)

## Acknowledgments

We thank Marissa Cruz for critical discussions and work on preliminary experimental data.

## Author Contributions

**Conceptualization:** Margaret L. DeMaegd, Wolfgang Stein.

**Data curation:** Margaret L. DeMaegd.

**Formal analysis:** Margaret L. DeMaegd.

**Funding acquisition:** Wolfgang Stein.

**Investigation:** Margaret L. DeMaegd.

**Methodology:** Margaret L. DeMaegd.

**Project administration:** Wolfgang Stein.

**Resources:** Wolfgang Stein.

**Software:** Wolfgang Stein.

**Supervision:** Wolfgang Stein.

**Validation:** Wolfgang Stein.

**Visualization:** Margaret L. DeMaegd, Wolfgang Stein.

**Writing – original draft:** Margaret L. DeMaegd, Wolfgang Stein.

**Writing – review & editing:** Margaret L. DeMaegd, Wolfgang Stein.

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
