## [Decision Letter · Decision Letter 0]

24 Mar 2020

Dear Dr. Stein,

Thank you very much for submitting your manuscript "Temperature-robust activity patterns arise from coordinated axonal Sodium channel properties." for consideration at PLOS Computational Biology.

As with all papers reviewed by the journal, your manuscript was reviewed by members of the editorial board and by several independent reviewers. In light of the reviews (below this email), we would like to invite the resubmission of a significantly-revised version that takes into account the reviewers' comments.

We cannot make any decision about publication until we have seen the revised manuscript and your response to the reviewers' comments. Your revised manuscript is also likely to be sent to reviewers for further evaluation.

Sincerely,

Joseph Ayers, PhD

Associate Editor

PLOS Computational Biology

Lyle Graham

Deputy Editor

PLOS Computational Biology

Reviewer's Responses to Questions

**Comments to the Authors:**

Reviewer #1: Uploaded as an attachment, "JB's comments on manuscript"

Reviewer #2: DeMaegd and Stein study how the pyloric rhythm produced by the stomatogastric ganglion can be preserved across a wide range of temperatures during transmission to muscles along axons. Using a preparation that allows measuring the conduction velocity in axons of different identified neurons at different temperatures, they find only mild temperature sensitivity of action potential velocity, resulting in a preserved phase relationship of the pyloric rhythm across temperatures. Additionally, they create a database of axon models with different temperature sensitivities of ion channels. Analysis of this database reveals relationships between parameters that allow preserving timing differences along axons across temperatures.

The authors perform nice experiments, but the finding that the pyloric rhythm is conserved in the axons across temperatures not very surprising. Further, the relationship between this main result and their modeling effort is not clear. Therefore, I cannot recommend this study for publication in its present form.

Major:

(1) The analysis of the relationship between different Q10s and timing differences between axons in Fig. 4 is nice, but the relationship to the experimental data is not clear. Is it possible to transform timing into phase differences and ask what combinations of Q10s result in phase differences within the experimentally determined range (see Fig. 2B)? If there is a large variety of Q10 combinations (even with some ‘coordination’) within this range of phase differences, this would be in line with ideas that in the CPG itself many different combinations of ion channel expression can result in a consistent pyloric rhythm – and that this principle also holds true for the transmission line to the periphery.

(2) The model description is incomplete and some key assumptions are left unexplained. It is a compartmental HH-type model of a 4 cm long axon with a diameter of 3 µm (I assume from Fig. 4, but this isn’t mentioned anywhere). The number of compartments is not relevant, rather the length of an individual compartment (50 µm?). Also missing in the Methods is the value of the axial resistance.

More importantly, why were axon diameters of 3, 6 and 12 µm used instead of the measured diameters? This might help directly relating the simulation results to the experimental measurements (see above). Finally, it is an unusual idea to assume a temperature-dependence of the maximum conductance density. Since it is a model parameter, it certainly is fair game, however I am not aware that this is necessary for a HH description of Na and K channels. At the end of the discussion, the authors seem to imply that this is a phenomenological approximation of possible neuromodulatory influence on e.g. channel availability at different temperatures. If this is the case, please clarify this much earlier (and provide evidence if possible). Also, is an exponential temperature dependence an appropriate assumption in this case?

Minor:

-The temperature dependence of the conduction velocity does not seem to follow a single exponential (Fig. 2Ai and 2Aiii); there seem to be two different regimes (from 5-15 C and from 20-25 C). This poses a problem for the claim that the Q10 values are significantly different, since this assumes a single exponential. Is it possible to come up with a different phenomenological description (e.g., two Q10 values for the two ranges)? And do the axon models display this phenomenon, and could they be used to explain it?

-lines 127-128: The extracellular AP amplitude indeed seems to be characteristic for each axon, but that does not necessarily mean that the diameter difference is consistent everywhere (the amplitude could also be influenced by different ion channel densities).

-lines 280-281: Unclear – maybe: ‘the temperature dependence of action conduction velocity is robust across a wide range of Q10s of the underlying ion channels’ or similar.

-Please replace ‘preciseness’ by ‘precision’ throughout.

-Fig. 4B: Please show examples of the data points with the linear regression to give an idea of how strong this relationship is

-Switch the rows in Fig. 4C to match how they are referenced in the text

-lines 361-370: Please provide some form of significance test for these statements.

-lines 436-446: This is discussion, not a result.

-lines 449-470: should not be part of discussion. Remove or integrate into introduction.

-lines 546-547: Please clarify/remove. I don’t think Q10s of individual ion channels can be ‘regulated’ through gene expression.

-lines 552-573: Seems very redundant with the results section.

-line 704: I think the ln in the exponent needs to be a log10; i.e., R = Q10 ^ ((T-10)/10).

Reviewer #3: This manuscript by DeMaegd & Stein addresses the question of the role of temperature in the conduction properties along the axon and activity patterns in the pyloric network of the crab stomatogastric system. In particular, the authors show that pyloric axons display variations in conduction velocity that are comparable over different types of axons (PD, LP, PY) yet displaying different conduction velocities and diameters. They next analyzed the reasons for this stability and found using a model that to maintain conduction timing similar amongst axon of different diameter, the sodium activation time constant (taum) and maximum sodium conductance (GNa) temperature sensitivity must be coordinated.

The results reported in this manuscript are interesting, the paper is well written but the presentation of some modeling results (see point 1 below) is hard to follow. In addition, the few points below should be addressed.

1. The color coding and data representation used in Figure 3B&C and 4A are really difficult to follow. The authors should consider a simplification of the message in both figures to allow the reader to easily extract pertinent information.

2. It is concluded in the abstract and elsewhere in the paper that “the effects of temperature on action potential propagation were subtle”. This is not exactly true since in Figure 2A, the authors show a two-fold increase in velocity when the temperature goes from 5 to 20°C (i.e. Q10 ~1.5). Please, amend the conclusion in the abstract and throughout the paper accordingly.

3. Figure 1B, the green axon has the same diameter as the red one. Perhaps, choose another region where the difference is not ambiguous.

4. Please specify the used animal species (crab) in the abstract and in the results.

5. Line 50, please insert a space (“, the” and not “,the”).

6. Page 18, line 355, add “between” after than.

7. Page 19, line 375, 0.375 and not .375

8. Page 20, line 394, suppress the dot after “and”

**Have all data underlying the figures and results presented in the manuscript been provided?**

Reviewer #1: Yes

Reviewer #2: Yes

Reviewer #3: Yes

PLOS authors have the option to publish the peer review history of their article (what does this mean?). If published, this will include your full peer review and any attached files.

Reviewer #1: No

Reviewer #2: No

Reviewer #3: No
---

## [Decision Letter · Decision Letter 1]

15 Jun 2020

Dear Dr. Stein,

We are pleased to inform you that your manuscript 'Temperature-robust activity patterns arise from coordinated axonal Sodium channel properties.' has been provisionally accepted for publication in PLOS Computational Biology.

Best regards,

Joseph Ayers, PhD

Associate Editor

PLOS Computational Biology

Lyle Graham

Deputy Editor

PLOS Computational Biology

Reviewer's Responses to Questions

**Comments to the Authors:**

Reviewer #1: The authors have done a very complete complete job of addressing my and the other reviewer's comments. The revised manuscript is considerably clearer and more accessible to the reader.

My few comments are of the trivial, editorial variety:

Line 29. Suggest adding comma after "restricted".

Line 57. Suggest adding comma after "behavior".

Line 151. Should "lvn_low" be italicized?

Line 181. Suggest changing" to "above".

Line 345. Suggest adding comma after "Q10s".

Line 413 and Fig. 4C. I believe "coefficient of variation" is typically abbreviated using capital letters (CV).

Line 527. I do not think "The" should be capitalized.

Line 583. Suggest adding "in stomatogastric motor neurons" after "5-fold".

Line 600. Suggest changing "the ion channel attain" to "associated with properties of the various ion channels" or "associated with the ion channels".

Line 614. Delete comma after "although".

Line 619. Suggest adding comma after "changes".

Lines 620, 622. Does "capacity" mean "capacitance"?

Line 624-626. Sentence starting with "Quicker" - I think there is a missing word. Add "and" before "indicates"? Or change "indicates" to ", indicating".

Line 630. Suggest deleting "most".

Reviewer #2: The revised manuscript is much improved and the authors have addressed all my concerns. My only remaining suggestion is to reference Figure 3C somewhere in lines 323-332.

Reviewer #3: The paper has been adequately revised. I have no further comment.

**Have all data underlying the figures and results presented in the manuscript been provided?**

Reviewer #1: Yes

Reviewer #2: Yes

Reviewer #3: Yes

PLOS authors have the option to publish the peer review history of their article (what does this mean?). If published, this will include your full peer review and any attached files.

Reviewer #1: No

Reviewer #2: No

Reviewer #3: No

---

## [Editor Report · Acceptance letter]

15 Jul 2020

PCOMPBIOL-D-19-01983R1 

Temperature-robust activity patterns arise from coordinated axonal Sodium channel properties

Dear Dr Stein,

I am pleased to inform you that your manuscript has been formally accepted for publication in PLOS Computational Biology. Your manuscript is now with our production department and you will be notified of the publication date in due course.

With kind regards,

Matt Lyles
